# Diabetes Mellitus: A Path to Amnesia, Personality, and Behavior Change

**DOI:** 10.3390/biology11030382

**Published:** 2022-02-28

**Authors:** Rahnuma Ahmad, Kona Chowdhury, Santosh Kumar, Mohammed Irfan, Govindool Sharaschandra Reddy, Farhana Akter, Dilshad Jahan, Mainul Haque

**Affiliations:** 1Department of Physiology, Medical College for Women and Hospital, Dhaka 1230, Bangladesh; rahnuma.ahmad@gmail.com; 2Department of Pediatrics, Gonoshasthaya Samaj Vittik Medical College and Hospital, Dhaka 1344, Bangladesh; konaonu96@gmail.com; 3Department of Periodontology and Implantology, Karnavati School of Dentistry, Karnavati University, 907/A, Uvarsad Gandhinagar, Gujarat 382422, India; santosh@ksd.ac.in; 4Department of Forensics, Federal University of Pelotas, Pelotas 96020-010, RS, Brazil; irfan_dentart@yahoo.com; 5Department of Periodontics and Endodontics, School of Dental Medicine, University at Buffalo, Buffalo, NY 14214, USA; sharasch@buffalo.edu; 6Department of Endocrinology, Chittagong Medical College, Chattogram 4203, Bangladesh; fakter36@gmail.com; 7Department of Hematology, Asgar Ali Hospital, 111/1/A Distillery Road, Gandaria Beside Dhupkhola, Dhaka 1204, Bangladesh; dilshad@asgaralihospital.com; 8Unit of Pharmacology, Faculty of Medicine and Defence Health, Universiti Pertahanan Nasional Malaysia (National Defence University of Malaysia), Kem Perdana Sungai Besi, Kuala Lumpur 57000, Malaysia

**Keywords:** diabetes mellitus, vascular dementia, Alzheimer’s Disease, inflammation, atherosclerosis, mitochondrial dysfunction, cognitive dysfunction

## Abstract

**Simple Summary:**

Diabetes Mellitus (DM) is a metabolic disorder resulting from a disturbance of insulin secretion, action, or both. Hyperglycemia and overproduction of superoxide induce the development and progression of chronic complications of DM. The impact of DM and its complication on the central nervous system (CNS) such as dementia and Alzheimer’s Disease (AD) still remain obscure. In dementia, there is a gradual decline in cognitive function. The incidence of dementia increases with age, and patient become socially, physically, and mentally more vulnerable and dependent. The symptoms often emerge decades after the onset of pathophysiology, thus impairing early therapeutic intervention. Most diabetic subjects who develop dementia are above the age of 65, but diabetes may also cause an increased risk of developing dementia before 65 years. Vascular dementia is the second most common form of dementia after AD. Type 2 DM (T2DM) increases the incidence of vascular dementia (since its covers the vascular system) and AD. The functional and structural integrity of the CNS is altered in T2DM due to increased synthesis of Aβ. Additionally, hyperphosphorylation of Tau protein also results from dysregulation of various signaling cascades in T2DM, thereby causing neuronal damage and AD. There is the prospect for development of a therapy that may help prevent or halt the progress of dementia resulting from T2DM.

**Abstract:**

Type 2 diabetes mellitus is increasingly being associated with cognition dysfunction. Dementia, including vascular dementia and Alzheimer’s Disease, is being recognized as comorbidities of this metabolic disorder. The progressive hallmarks of this cognitive dysfunction include mild impairment of cognition and cognitive decline. Dementia and mild impairment of cognition appear primarily in older patients. Studies on risk factors, neuropathology, and brain imaging have provided important suggestions for mechanisms that lie behind the development of dementia. It is a significant challenge to understand the disease processes related to diabetes that affect the brain and lead to dementia development. The connection between diabetes mellitus and dysfunction of cognition has been observed in many human and animal studies that have noted that mechanisms related to diabetes mellitus are possibly responsible for aggravating cognitive dysfunction. This article attempts to narrate the possible association between Type 2 diabetes and dementia, reviewing studies that have noted this association in vascular dementia and Alzheimer’s Disease and helping to explain the potential mechanisms behind the disease process. A Google search for “Diabetes Mellitus and Dementia” was carried out. Search was also done for “Diabetes Mellitus”, “Vascular Dementia”, and “Alzheimer’s Disease”. The literature search was done using Google Scholar, Pubmed, Embase, ScienceDirect, and MEDLINE. Keeping in mind the increasing rate of Diabetes Mellitus, it is important to establish the Type 2 diabetes’ effect on the brain and diseases of neurodegeneration. This narrative review aims to build awareness regarding the different types of dementia and their relationship with diabetes.

## 1. Introduction

Diabetes Mellitus (DM) is a metabolic disorder resulting from a disturbance of insulin secretion, action, or both. There has been an increase in individuals suffering from DM—108 million in 1980 to 422 million in 2014. This value is expected to rise to 693 million by 2045 [1,2,3].

Hyperglycemia and overproduction of superoxide induce the development and progression of chronic complications of DM [4,5,6]. The significant pathways of diabetic chronic complications include protein kinase activation, advanced glycosylation end-product, inflammation, expression and action of cytokines, inflammatory mediators and hormones, polyol pathway, and increase in hexosamine activity [7,8]. There is a less clear understanding of the impact of DM and its complication on the central nervous system [9]. An association has been observed between DM and declines in cognition with increased risk of development of dementia, including vascular dementia and Alzheimer’s Disease (AD) [10].

Dementia is a dangerous disease with a progressive decline in cognition. Its incidence increases with age, and those with dementia become socially, physically, and mentally more vulnerable and dependent. Symptoms may emerge decades after the onset of pathophysiology, thus hampering disease-targeted therapy [11].

Worldwide, about 4.6 million cases of dementia occur every year. The number of individuals suffering from cognitive decline is expected to double every 20 years [12]. The increase in risk of developing dementia in diabetes varies with age, education, ethnicity, macrovascular and microvascular diseases, presence of depression, lower extremity complications, and diabetes of longer duration [13,14,15].

It has been reported that vascular dementia is the second most common form of dementia after AD [16,17]. Type 2 diabetes increases the incidence of vascular dementia since it impacts the vascular system. Patients with Type 2 Diabetes may also have AD [18]. Table 1 includes studies linking structural and functional changes in the nervous system in dementia subjects with DM.

In dementia, there is a gradual decline in cognitive function. Most diabetic subjects who develop dementia are above the age of 65, but diabetes may also cause an increased risk of developing dementia before 65 years [19,20,21]. A meta-analysis in considering prospective observational studies regarding DM and risk of dementia reported that the relative risk for dementia was noted to be 1.73 (1.65–1.82) and for vascular dementia to be 2.27 (1.94–2.66) for diabetic subjects [22]. Another meta-analysis of cohort studies on DM and risk of Alzheimer’s Disease observed a relative risk of AD—1.53 (1.42–1.63)—for diabetic individuals [23].

**Table 1 biology-11-00382-t001:** Studies Linking Diabetes Mellitus to Dementia.

Reference	Study Population	Study Design	Study Period	Subgroup	Results
Espeland MA et al. 2013 [24]	N = 1366 Age range = 72–89 years at start of study; 698 women obtained a repeat MRI scan	Cohort study	4.7 years	Women with DM and women without DM	145 diabetic women had smaller brain volume (0.6% less; *p* = 0.05), gray matter volumes that was smaller (1.5% less; *p* = 0.01), ischemic lesion volumes which was larger (21.8% greater; *p* = 0.02), in white matter (18.8% greater; *p* = 0.02), both overall and in gray matter (27.5% greater; *p* = 0.06).
Moran et al. 2013 [25]	N = 713 Age = ≥55 years	Cross-sectional study	2 years (2008–2010)	Subjects with Type 2 DM (T2DM) = 350 Subjects without Type 2 DM (T2DM) = 363	An MRI scan revealed an association betweenT2DM and greater cerebral infarcts and lesser total white, gray, and hippocampal volumes (*p* < 0.05). T2DM was associated with poorer visuospatial construction, planning, speed, and visual memory (*p* ≤ 0.05)
Ball et al. 2011 [26]	N = 4 tissue culture and brain slices from streptozotocin (STZ)-diabetic rats	Experimental animal study	2–3 weeks	Relative connexin (Cx) protein levels were assessed by Western blotting using extracts from cultured astrocytes grown in high (25 mmol/L) or low (5.5 mmol/L)	Astrocytic growth in high glucose reduced dye-labeled area by 75%; actin level rose by 24%. Oxidative stress and regionally-selective down-regulation of connexin protein content affect gap junctional communication in brain of STZ-diabetic rats
Janelidze et al. 2017 [27]	N = 1015 Age = ≥60 years	Cohort study	5.7 yrs (3.0–9.6)	Alzheimer’s Disease, Vascular dementia, dementia with Lewy bodies (DLB), Parkinson’s disease with dementia (PDD), and frontotemporal dementia (FTD)	The CSF/plasma albumin ratio (an indicator of BBB and blood-CSF barrier permeability) was increased in individuals with diabetes (diagnosed with diabetes or taking antidiabetic medications) compared with those without diabetes (*p* = 0.015). Diabetes was linked to high CSF levels of ICAM-1 (*p* < 0.001), VCAM-1 (*p* = 0.007), and VEGF (*p* = 0.024).
Navaratna et al. 2013 [28]	Diabetes induced rats using streptozotocin	Experimental animal study	6–12 weeks	Human brain microvascular endothelial cells with Advanced Glycation endproduct-BSA ((0–200 μg/mL) and treatment with non-glycated BSA (100 μg/mL)	Treatment with AGE-BSA (0–200 μg/mL) induced a dose-dependent increase in endothelial levels of MMP9. Study also showed neuronal TRKB trophic function is processed by MMP9-mediated degradation in the diabetic brain.
Vergoossen et al. 2020 [29]	N = 2302; age: 59 ± 8 years	Cohort study	4 years (2013–2017)	1361 subjects with normal glucose metabolism, 348 with prediabetes, and 510 with type 2 diabetes	Association of prediabetes and type 2 diabetes with white matter network organization was studied. Prediabetes and type 2 diabetes were linked with a lower node degree after full adjustment (standardized [st]β_Prediabetes_ = −0.055 [95% CI −0.172, 0.062], stβ_Type2diabetes_ = −0.256 [−0.379, −0.133], *P*_trend_ < 0.001). Prediabetes was associated with lower local efficiency (stβ = −0.084 [95% CI −0.159, −0.008], *p* = 0.033) and lower clustering coefficient (stβ = −0.097 [95% CI −0.189, −0.005], *p* = 0.049).
Jackson et al. 2013 [30]	N = 56; Age = >70 years	Post mortem study		Brain tissues were divided in pathologically distinct groups: the group of brain tissues from patients with overt Type2DM and dementia (N = 15); brain samples from Alzheimer’s Disease patients without history of Type2DM (N = 14); brain specimens from age-matched healthy individuals as controls (N = 13)	Amylin oligomers and plaques were noted in the temporal lobe gray matter from patients with diabetes, but not found in controls. In addition, amylin deposit was observed in blood vessels and perivascular spaces.
Currais et al. 2012 [31]	Type 1 Diabetes induced mice; mean age = 6 months	Experimental animal study	4 months	Type1 DM induced senescence-accelerated prone 8 (SAMP8) and senescence-resistant 1 (SAMR1) mice. Age matched non-diabetic SAMP8 mice	Type 1 Diabetes Mellitus increased Aβ and glial fibrillary acidic protein (GFAP) immunoreactivity in the hippocampus of SAMP8 mice and in age-matched SAMR1 mice to a lesser extent. Analysis showed aggregation of Aβ within astrocyte processes surrounding vessels. Western blot analyses from Type 1 Diabetes Mellitus SAMP8 mice showed raised APP processing and protein glycation along with increased inflammation. Type 1 Diabetes Mellitus increased tau phosphorylation in the SAMR1 mice but did not further increase it in the SAMP8 mice
Willette et al. 2015 [32]	N = 186; age = 60.37 ± 5.63	Cross-sectional study		Normoglycemic 135 Pre Diabetic 43 Diabetic 8	Pittsburgh Compound B (PiB) Positron Emission Tomography revealed in participants with normoglycemia, higher insulin resistance corresponded to higher PiB uptake in frontal and temporal regions, suggesting raised amyloid deposition.

## 2. Objectives of the Study

The study’s objective was to review human and animal studies to understand the relationship between DM and dementia, considering its relationship with vascular dementia and AD individually. An attempt was made to relate the possible mechanisms leading to dementia in diabetic individuals to build awareness regarding this life-altering debilitating condition of the brain as a comorbidity of DM.

## 3. Material and Methods

This narrative review focuses on identifying the relationship of DM with dementia and the possible pathology leading to dementia. The study was carried out between October and December 2021. The search was carried out from an electronic database using Google search engine, Google Scholar, Science Direct, PubMed, MEDLINE, and Embase. Related articles from the list of references were searched to obtain more articles on the topics. Keywords used in search of related articles were “Diabetes Mellitus”, “Dementia”, “Vascular Dementia in Diabetes Mellitus”, “AD”, “Diabetes Mellitus and AD”, “Inflammation in Dementia”, “Neurodegeneration”. Articles and literature dating before 2000 and articles unavailable in English were excluded from the review. A hand-search of relevant articles was carried out before inclusion in this study.

## 4. Vascular Dementia

Vascular Dementia is characterized by the reduced flow of blood to the brain, affecting cognitive function, especially execution. Individuals with vascular dementia suffer from forgetfulness, anxiety, depression, loss of function like working memory, reasoning, planning, task execution, and thinking. About 17–20% of dementia patients suffer from vascular dementia [33,34,35]. DM is a risk factor for vascular dementia [34,36,37,38,39].

## 5. Pathology of Vascular Dementia

The pathology of vascular dementia involves both large and small blood vessels. Development of microinfarct, Lacunar infarct, macro infarct, micro-bleed, and white matter changes are observed in patients suffering from this type of dementia [40]. Microinfarcts and microhemorrhages are associated with pathologies of blood vessels like lacunar infarct (small infarcts of white matter, especially in basal ganglia), large infarcts, leukoaraiosis, and hemorrhage [41,42,43] (Figure 1). Microinfarcts are lesions composed of necrosis, inflammation, cavitation, and palor with infiltration of microglia, macrophage, and astrocyte [44]. Microbleeds in the deep region of the brain are associated with white matter that occurs secondary to vascular risk factors [45,46]. As observed in neuropathological studies, lacunar infarcts and microinfarcts are important risk factors for developing pure vascular dementia [40].

Vascular lesions that lead to vascular dementia include atherosclerotic plaques affecting small cerebral vessels, deposition of hyaline substance in the vessel wall (lipohyalinosis), distortion of microvasculature, vessel wall stiffening due to fibrosis, and vessel wall integrity loss [44]. Vascular lesions that have been noted are tortuous arterioles with the thickened basement membrane, nonfunctional capillaries with no endothelial cells, and venules having collagen deposits [47]. Such lesions result in demyelination, axon loss, vacuolation, and lacunar infarcts, thus damaging the white matter [48]. Impairment of cognition correlates with white matter lesion expansion having new lacunes leading to sharper decline, particularly in executive and motor function [49].

## 6. Diabetes Mellitus as a Risk Factor for Vascular Dementia

DM acts as a risk factor for vascular changes in the brain that lead to vascular dementia [40]. MRI scan of the brain in elderly diabetic patients with no history of stroke revealed silent brain infarctions, cerebral microbleeds, and white matter lesions [50,51]. Neuroimaging of diabetic subjects also showed lacunar infarcts and brown atrophy [24,25]. Cerebral large vessel diseases in diabetic individuals include carotid artery disease and intracranial artery disease [41]. Large vessel infarction can result from atherosclerotic stenosis that causes large vessel occlusion or critical distal flow impairment [40,52]. An environment of inflammation created in chronic metabolic conditions like DM contributes to pathological changes in vasculature in the human body [53].

Vascular dysfunction was reported in subjects with uncontrolled Type 2 DM [54]. DM and its risk factors of blood vessels may result in hypoperfusion of the cerebrum [10,55]. A study that used non-ionizing Arterial Spin Labeling Magnetic Resonance Imaging (ASL-MRI) observed regional hypoperfusion of cerebrum (like in visual and cerebellar network) in diabetic subjects when compared to non-diabetic subjects [56]. Another study performed in 2018 found hypoperfusion in the frontal, inferior parietal, inferior temporal cortices, and hippocampus [57]. Several studies have found a correlation between hypoperfusion of the cerebrum and decline of cognition [56,57,58,59].

## 7. Inflammation of Blood Vessels in Diabetes Mellitus

In blood vessels of both the periphery and central nervous system, there are advanced glycation end products (AGE) from blood protein glycation resulting from hyperglycemia in DM. Accumulation of AGEs may lead to inflammation of vasculature through interaction between AGE and receptor for AGE (RAGE) [60,61]. AGE and RAGE interaction result in upregulation of vascular cell adhesion molecule 1 (VCAM-1) and activation of NF kβ. VCAM 1 enhances the adhesiveness of monocyte permeability of vasculature while production of NF kβ promotes proinflammatory and atherosclerotic changes in vascular endothelium and smooth muscle cells [62,63].

## 8. Inflammation, Atherosclerotic Change in the Blood Vessel of Diabetics

Risk factors for atherosclerosis formation in diabetic patients include chronic hyperglycemia, hyperinsulinemia, dyslipidemia, and hypertension [64]. In a state of hyperglycemia, activation of the polyol pathway, protein kinase C, and production of AGE results in cell damage [65]. There is a reduction in endothelial nitric oxide synthase activity and decreased nitric oxide production, which causes endothelial dysfunction [66]. Atherosclerosis is promoted with eventual thrombus formation due to increased adhesion molecule expression in the endothelium, reduction in vasodilation, and inflammatory action. These changes ultimately may lead to cerebral infarction [50,67].

Reactive Oxidative Species (ROS) production increases in DM [68]. A link between hyperglycemia and increased ROS formation was observed in studies carried out in vitro [69]. An increase in intracellular blood glucose may alter metabolic pathways like the cell’s electron transport system, resulting in ROS overproduction. Metabolites of glucose also activate aldose reductase and protein kinase C-beta causing inflammation [70].

The increased formation of AGE molecules in a hyperglycemic state causes adhesion molecules to become activated, increasing monocyte or macrophage adhesion and entry into the sub-endothelium at the beginning of plaque formation. Macrophages release increased cytokines under AGE’s influence, thus maintaining proinflammatory conditions for plaque development. AGE also aggravates the development of atherosclerosis by causing excessive glycation of the extracellular matrix protein and promoting interaction with RAGE on endothelial cells macrophages. Such activity leads to proinflammatory conditions and excessive ROS in the cells [71,72,73]. An acceleration of atherogenesis, with macrophage infiltration, enhanced inflammatory markers expression, and increased plaque size was noted in diabetes-induced mice that were glutathione peroxidase 1 deficient [74].

Endothelial dysfunction is marked by inflammation cascade and monocyte infiltration during atherosclerotic plaque formation, which changes macrophage. The macrophage internalizes the low-density lipoprotein (oxidized), which converts to foam cells. The dead foam cells remodel and rupture, resulting in thrombus formation and occlusion of the vessel [75].

## 9. Altered Blood–Brain Barrier Integrity in Vascular Dementia

Blood–Brain Barrier (BBB) permeability alterations have been associated with lacunar stroke and leukoaraiosis [76,77]. In vascular cognitive impairment, plasma protein albumin was noted to be increased in Cerebrospinal Fluid (CSF), suggesting BBB breakdown [78]. Molecular alterations resulting in dysfunction of the BBB may include plasma protein transcytosis due to enlargement of caveolae of the endothelium [79,80], increase in metalloprotease expression [81], and reduced junctional adhesion and tight junction protein [82]. Brain endothelial penetrability is increased due to inflammatory agents. The bradykinin receptor, when activated in endothelial cells, leads to a rise in the concentration of intracellular calcium ions [83,84] with endothelial nitric oxide synthase activation. This causes the opening and increased permeability of tight junctions [85]. This effect is further aggravated with the release of IL 6 from astrocytes when Bradykinin activates the NF-*κ*B pathway within the astrocytes, and TNF-α deepens BBB permeability by acting directly on the endothelium and also through the production of endothelin-1 and release of IL-1β from astrocyte [86,87]. The release of IL-1β may result in a reduced concentration of tight junction protein called occludin and, therefore, increase the BBB’s permeability [88]. In Type 2 DM, primary sources for inflammatory cytokines are IL-1β, IL-6, and TNF-α; the process is activated by the macrophages found in adipose tissue [89]. Cytokines leak from the blood into brain parenchyma through regions lacking BBB (circumventricular organs). These cytokines may cause macrophage activation, thus inducing a cascade of pro-inflammatory changes. In the case of massive activation of neurons, many metalloproteases result in BBB breakdown [90].

Studies observing the effect of increased blood glucose levels on astrocytes in humans noted significantly increased production of inflammatory cytokines like TNF α, IL 1, IL 4, IL 6 utilizing STAT 3 and NF kβ pathways of inflammation [91]. In brain slices and tissue culture of diabetic rats, astrocyte gap junction communications were seen to be inhibited. There was also an overproduction of reactive oxygen and nitrogen species [26,92]. There has also been a report of an increase in VEGF (vascular endothelial growth factor) under the influence of AGE [93]. VEGF increases BBB permeability by increasing and promoting GLUT 1 translocation to the cell surface and reducing inter-endothelial tight junction proteins like occludin and ZO-1 [92]. A study of human brain microvascular endothelial cells to compare the effects of TNF α and IL 6 on BBB characteristics noted a significant decrease in all inter endothelial junctional proteins (Occludin, Claudin-5, and VE-cadherin) and an increase in endothelial permeability [94]. A study has also found a correlation between increased BBB permeability and dementia development [27]. Matrix metalloproteinase (MMP) plays a significant role in altering BBB in hyperglycemia [95]. AGE molecules have been noted to promote MMP-2 release [93].

In a hyperglycemic state, glucose utilization occurs through protein kinase C and AGE pathways, which cause overproduction of superoxide [96]. Activation of protein kinase C leads to zonula occludens-1 protein (ZO 1) phosphorylation, disruption of Tight junction, and increased expression of VEGF [97]. AGE molecules interact with the integrin of the cell membrane and AGE receptor. Activated RAGE increases ROS production with the eventual activation of NF κβ, which promotes the release of inflammatory mediators [98,99,100].

## 10. Inflammation Oxidative Stress in Diabetes Mellitus Linked to Vascular Dementia

Insulin resistance in DM has been observed to cause oxidative stress and inflammation in both humans and animal models [101,102]. Nicotinamide adenine dinucleotide phosphate hydrogen (NADPH) oxidase is an important cause of vascular oxidative stress in diabetes [103]. The NADPH oxidase enzyme complex activates monocytes and causes superoxide anion to explode (O_2_^−^). Monocyte-derived superoxide anion synthesis comes up with atherosclerotic formation by switching smooth muscle cell mitogenic signaling pathways [104,105,106]. The ROS-generating NADPH oxidase enzyme definitely correlated with the pathogenesis of several neurodegenerative diseases and neuroinflammation, including dementia [107,108]. AGE is formed in diabetic subjects, resulting in secretion of MMP-9 from endothelial cells and brain-derived neurotrophic factor (BDNF) receptor and also cleavage tyrosine kinase receptor B (TrkB), thus decreasing neurotrophin signaling [28]. Neurons and glia provide trophic support to vascular cells, and therefore damage of these supporting cells results in atrophy of endothelial cells and rarefaction of microvasculature [47,109]. Loss of myelin sheath’s normal physiology and anatomy, including demyelination of axons, has been reported as one of the dangerous outcomes of the inflammatory process and ROS [110,111]. Demyelination leads to disruption of the integrity of axons exposure to the damaging effects of free radicals and cytokines in the brain’s white matter [112,113]. Microtubule fragmentation and disruption of axon flow eventually occurs as the Na^+^/K^+^ ATPase pump fail in the axon with an accumulation of intracellular calcium that activates activities that are protease dependent [113,114].

Damage to the white matter can affect the fidelity and precision of the transfer of information for brain functioning and cognition [115,116]. The myelinated tracts of white matter serve the function of long-range connectivity, synchronization between hemispheres, and neurotrophic activity through axon flow and plasticity [41,116,117]. Lesions of white matter thus affect the structure and function of the brain with a decrease in utilization of glucose by the frontal lobe [41,118] and disruption of brain connectivity [29,119,120]. Damage to myelin sheath may compromise skilled motor learning and neuroplasticity functions, thus leading to impairment of cognition [41].

## 11. Alzheimer’s Disease

Alzheimer’s Disease (AD), the most common form of dementia, is a disorder of the brain that is irreversible and progressive and accounts for about 60–70% of all dementia cases [11,121,122]. There is a rapid rise in the prevalence of individuals with AD in the age group 65 years and above. In 2020, it was reported that about 5.8 million people were living with AD in America. A recorded death of 122,019 patients in 2018 in America makes it the sixth leading reason for death in the United States of America [123].

Symptoms of the disorder may be observed following changes in the brain resulting from destruction or damage to neurons of areas of the brain related to functions of cognition like learning, memory, and thinking [30,122]. Symptoms to appear at an early stage of the disorder are depression and impairment of cognition with early presentation of memory loss as early as 12 years before the onset of clinically defined AD. There are subsequent symptoms of behavior deficit, language deficit, disorientation, and psychosis. Patients develop myoclonus, disturbed gait, and rigidity [124]. In severe disease cases, the patients become bedbound and experience difficulty swallowing when the brain area concerned with swallowing is damaged. This can result in food entering the trachea and eventually causing aspiration pneumonia, leading to the patient’s death [125,126].

## 12. Pathogenesis of Alzheimer’s Disease

AD is characterized by:(1)The formation of senile plaque, which is an extracellular lesion consisting of an accumulation of amyloid-β (Aβ) protein-42 (Aβ42) in its nucleus.(2)Neuro fibrillar tangles that are intraneuronal findings consisting of phosphorylated tau protein (P-tau) [11].

There is brain protein misfolding with deposition of extracellular amyloid plaque followed by neurofibrillary tangles deposition and neuronal death in the brain [127,128,129]. Accumulation of β amyloid protein in the capillary wall, arteries, and arterioles causes cerebral amyloid angiopathy with eventual degeneration of vascular wall components and impairment of blood flow. Such accumulation also causes inflammation of astrocytes, microglia, and the central nervous system [129].

Aβ protein (36–43 amino acid containing peptide) is a component of Amyloid Precursor Protein (APP), a transmembrane protein, and arises when APP is cleaved by β and γ secretase. APP cleavage by β secretase at N terminal results in APP C terminal fragment formation, which is then cleaved by γ secretase to form Aβ [130]. During the process of cleaving APP, when there is a defect in the clearance of β amyloid protein, insoluble Aβ accumulates [131]. Soluble oligomers initially form from the polymerization of the monomer of Aβ. Then further polymerization leads to the production of large fragments such as Aβ42, which then form insoluble amyloid fibrils [122,132].

Aβ reduces metal ions to produce H_2_O_2_ (hydrogen peroxide) and also increases the production of free radicals with zinc, copper, and iron, which is concentrated in both periphery and core of deposits of Aβ [122,133]. Excessive Ca^2+^ in the cytosol due to stored Ca^2+^ depletion in the endoplasmic reticulum may occur due to Aβ plaque formation [134]. A rise in cytosolic Ca^2+^ causes a fall in endogenous glutathione levels and accumulation of ROS within the cell [135]. Stress/c-Jun N-terminal kinase (JNK) activated protein kinase pathways are promoted by ROS, resulting in hyperphosphorylation of tau protein [136]. Aβ also activates NADPH (oxidase pathway, therefore, encourages the formation of free radicals, leading to excessive accumulation of ROS. ROS generated by Aβ can promote hyperphosphorylation of tau protein and alter cell signaling through MAPK (p38 mitogen-activated protein kinase) activation [137]. Figure 2 shows the pathogenesis of Alzheimer’s Disease.

Tau protein contributes to the stabilization of microtubules of axons [138]. In the brains of individuals with AD, there is hyperphosphorylation of tau protein that leads to the loss of the protein’s ability to bind to microtubules. There is thus disruption of microtubular stability and eventual death of neurons [139]. Neuro fibrillar degeneration can occur when paired helical filaments are formed from insoluble phosphorylated tau protein [140].

## 13. Diabetes Mellitus, Insulin Resistance, and Alzheimer’s Disease

A link between Type 2 DM and the risk of developing AD has been observed [11,40,141,142,143]. Mechanisms suggested for this link include insulin deficiency, insulin resistance, insulin receptor impairment, hyperglycemia, formation of advanced glycated end products, damage of cerebral vessels, and inflammation of vessels [32,144].

Insulin with insulin-like growth factors causes modulation of growth, differentiation, migration, and metabolism of neurons in the brain. They are also involved in gene expression, protein synthesis, formation, synapse plasticity, myelin production, and the maintenance of oligodendrocytes [145]. A bidirectional relationship exists between insulin resistance and AD [146]. In diabetic subjects, an inflammatory cytokine produced due to chronic peripheral inflammation can promote insulin receptor substrate-1 phosphorylation, thereby causing inhibition of signaling pathways downstream, such as JNK, I Kappa B kinase (IKK), and extracellular signal-regulated kinase 2 (ERK2), which may downregulate signaling mediated through insulin receptors and thus block insulin signaling [122,147].

In addition, systemic inflammation can cause damage to BBB and therefore trigger inflammation within the brain [148]. The increased rate of cell death, reduction of synapse function, neurogenesis inhibition, and death of neurons are promoted by circulating cytokines [149]. Transfer of Aβ to the periphery from CNS is inhibited by systemic inflammatory cytokines like TNFα, IL 6, and C reactive protein [150]. In turn, the accumulation of Aβ can cause the activation of microglia in the brain, which secrete inflammatory cytokines like IL6, IL1β, TNFα. These cytokines can bind to insulin receptors and activate IRS-1 serine kinase, which in turn phosphorylate IRS and thus cause alteration of insulin signaling in the brain [151,152].

Proinflammatory cytokines produced by microglia in the brain can also promote oxidative stress, which impairs insulin signaling, loss of synapse, reduction in mitochondrial transport in axon [153,154,155], fragmentation, and dysfunction of mitochondria [156] [Figure 3]. Dysfunction of mitochondria associated with a metabolic syndrome like diabetes, insulin resistance, and obesity has been an early change in AD [157]. Dysfunction of mitochondria includes impaired catabolism of substrate, dysregulated buffering of Ca^2+^, alteration of dynamics of mitochondria, and compromised transport of iron. Such dysfunctions result in inadequate ATP production, a rise in ROS, and cellular death [157,158,159]. There is the involvement of mitochondrial dysregulation of calcium ion signaling in the pathogenesis of insulin resistance and Type 2 diabetes [160]. In addition, alteration of Endoplasmic Reticulum-mitochondrial calcium ion signaling plays a significant role in the neurodegeneration development in the case of Alzheimer’s Disease [161,162].

Association between insulin signaling defect and AD has been found [163]. Severely diminished insulin receptor phosphorylation has been detected in the brains of subjects suffering from both diabetes and AD [164]. Such insulin signaling disturbance could promote an environment prone to metabolic stress in the CNS, which in turn cause dysfunction of the neuron [30,165]. In diabetic individuals, there is an accumulation of islet amyloid polypeptide (IAPP) within the islet of the pancreas, which is secreted alongside insulin. In patients with diabetes and AD, there is misfolding and elevation of IAPP [166] and accumulation of increased quantity of Aβ, tau hyper-phosphorylation [167]. AD and diabetic patients share atrophy of the brain, cerebral glucose reduction, and insulin resistance in the CNS [168].

Abnormal glucose metabolism also causes activation of a glycation reaction, which results in the production of AGE (advanced glycated end products). A rise in AGE in the brain and circulation has been connected to cognitive impairment in patients with Alzheimer’s [169]. Various studies have observed that an increase of AGE accumulation in diabetic rats’ brains suggests that removal of Aβ42 is impaired by AGE products and promotes aggregation of Aβ in the brain [167,170]. Figure 4 shows the different mechanisms in DM leading to AD.

## 14. Diabetes Mellitus, Inflammation, and Metabolism of Energy

Glucose metabolism in neurons comprises mechanisms regulating insulin, insulin signaling pathways, glucose transporters, and glycolytic end-product entry in mitochondria, which generates ATP through oxidative phosphorylation [171]. Mitochondria participate in metabolism signaling pathways like JNK, 5′AMP activated protein kinase signaling, redox-sensitive signaling, and cytosolic signaling. These signaling pathways and metabolite transporters, enzymes, and receptors ensure the neuron’s proper energy metabolism. One of the prime features of AD is an alteration of glucose metabolism in mitochondria marked by insulin signaling impairment, receptor activity alteration, and reduction in glucose uptake [172].

Immune cells infiltration from the periphery and activation of microglia cause the initiation of a cascade of intracellular signaling pathways that modify energy metabolism by mitochondria [173]. Impairment of oxidative phosphorylation is induced by inflammatory cytokines released from activated microglia. The activated microglia modulate astrocyte activity and deteriorates neuron integrity [174]. In the postmortem AD brain, pyruvate dehydrogenase activity was noted to be impaired, and elevated levels of IL1β, TNFα, and IL6 were observed, implicating hampering of tricarboxylic acid (TCA) cycle activity due to inflammation in AD patients and mild impairment of cognition [122].

A study on mouse hippocampal cell line and neuronal cell culture have observed that exposure to TNF α results in a decrease of ATP formation and basal respiration [175]. A reduction in peroxisome proliferator-activated receptor γ coactivator 1α (PGC1α), a regulator of biogenesis and function of mitochondria, in cardiomyocyte and myoblasts in human upon TNF α exposure have also been observed. However, such an effect has not yet been reported in neuronal cells in various neurodegeneration [122,176,177]. PGC 1α plays a vital role in different metabolic and cellular processes, including energy metabolism, neurodegenerative disease, and cardiovascular disease [178]. In the brain, a decrease in PGC 1α causes hyperactivity resulting from axonal degeneration within the brain [177].

Some mitochondrial vital enzymes such as α ketoglutarate dehydrogenase, Pyruvate Dehydrogenase, and enzymes of electron transport chain like cytochrome oxidase, are all reduced in the AD brain [179]. Studies of the AD brain have found that there is the oxidation of enzymes such as glyceraldehyde 3 phosphate dehydrogenase, α-enolase, fructose bisphosphate enolase, and enzymes involved in the Krebs cycle and glycolysis [180]. Metabolic functions are deranged due to inflammation resulting from these oxidative changes. Association between APP, Tau, Aβ, and impaired energy metabolism in mitochondria has been observed in studies [181,182]. Interaction of Aβ, hyperphosphorylated tau protein with mitochondrial protein has been suggested to disrupt the electron transport chain, and increase the production of superoxide radicals and ROS, which hinder the generation of ATP in cells [183]. It has also been suggested that the decline of mitochondria and other cellular organelles of synapses and terminals of nerves occurs due to the aggregation of Aβ and Tau. This may lead to severe depletion of ATP and starving of synapses and dendritic spines [184].

## 15. Dysfunctional Adipose Tissue, Diabetes Mellitus, and Dementia

Metabolic alterations associated with excess adipose tissue are possible risk factors for age-linked cognition decline. Inflammation along with oxidative stress are the factors that contribute to insulin resistance in the brain and the progress of cognitive decline [185]. In the case of excess adipose tissue, there is the release of excessive free fatty acid, which may lead to insulin resistance in the brain. The Free Fatty Acid may produce inflammatory cytokines, which stimulate serine kinases like IKK kinase and JNK. These events eventually inhibit the insulin receptor substrate 1(IRS1) signaling pathway by enhancing the phosphorylating the serine of IRS instead of tyrosine [145,186,187,188,189,190,191,192]. Excess Free Fatty Acid led to oxidative stress with overproduction of free radicles, which can damage mitochondria, lysosome, endoplasmic reticulum, and DNA [193]. Insulin resistance in the brain and oxidative stress eventually lead to AD [142,164,194].

Adiponectin is an adipocyte-derived factor that decreases when adiposity increases [195]. Low circulating adiponectin levels have been noted in heart failure, hepatic stenosis, dyslipidemia, and Type 2 DM [196]. Low levels of adiponectin have been associated with cerebrovascular disease [197]. A study noted subjects with cerebral infarction and atherosclerotic condition had lower adiponectin levels than those who were healthy [198]. Another case-control study found significantly lower adiponectin levels in patients with ischemic cerebrovascular disease [199]. The cerebrovascular disease has been a primary contributary factor for vascular cognitive impairment and, therefore, leads to vascular dementia [200].

## 16. Glymphatic System Disruption and Dementia

Glymphatic system dysfunction may contribute to the development of AD [201]. The glymphatic system lies between the astrocyte vascular end feet and vessel adventitia [202]. It acts as a waste product drainage system within the brain and delivers nutrients throughout the brain from CSF [203,204]. The system helps in cortical astrocyte Ca^2+^ signaling [205], norepinephrine regulation [206].

Glymphatic system dysfunction has been noted to aggravate AD symptoms [207,208]. In AD, the breakdown of BBB causes an increase of Aβ accumulation in plasma, interstitial fluid, and CSF [209], leading to synapse dysfunction within the brain. Disruption of BBB also results in inflammation, leading to dysfunction of the glymphatic system and impairment of glymphatic clearance [210,211].

About 60% of the Aβ of the brain is drained via the glymphatic system to lymph nodes [212]. Increased BBB permeability causes impairment of the glymphatic system and results in a defect of Aβ clearance by BBB [213,214]. Since in the brain of AD, patients cannot control the process of Aβ efflux and influx through the glymphatic system, Aβ accumulation takes place in the vascular structure and parenchyma of the brain. Type 2 DM aggravates BBB disruption and eventually triggers a decline in cognition through an imbalance of metabolites resulting from dysfunction of the glymphatic pathway [201].

## 17. APO Lipoprotein E: Type 2 Diabetes and Alzheimer’s Disease Relationship Modifier

Apolipoprotein E modulates the relationship between AD and Type 2 diabetes [215]. The allele APO E ε 4 is the most important genetic risk factor for late-onset AD. ApoE takes part in the transport of lipid, metabolism of lipoprotein, and regulation of repair of neurons, the genesis of the synapse, nerve development, and growth [216]. Studies have observed in the presence of the allele Apo E ε 4, there is a rise in the deposition of Aβ and thus influences the AD pathology [217].

Type 2 diabetic patients with ApoE ε 4 carriers have an increased risk of developing AD [218]. There is an increase in neurofibrillary tangles, formation of amyloid plaque, and cerebral amyloid angiopathy in the presence of this allele in Type 2 Diabetic subjects with AD. Insulin degrading enzymes are lower in carriers of ApoE ε 4, which alter insulin signaling and clearance of Aβ in Type 2 Diabetes and AD [219,220]. Insulin level in plasma and cerebrospinal fluid is higher in AD subjects who were non-carriers of allele ApoE ε 4 [221,222,223]. In addition, administration of insulin nasal spray was observed to have a more positive effect on memory, and pathology of Aβ are affected by Apo E genotype in AD subjects [224,225]. Brain peroxisome proliferator-activated receptor γ and its coactivator PGC1α has a role in regulating glucose uptake. The metabolism was found to be downregulated in APOE4-TR mice when compared to APOE3-TR mice [226]. Other studies have also found a decrease in glucose uptake and impairment of insulin signaling in APOE4-TR mice compared to APOE3-TR mice [227,228]. APOE4 impairs insulin signaling by interacting with the insulin receptor and entrapping them within endosomes. Such studies indicate that the genotype of ApoE worsens AD in Type 2 diabetic subjects [216].

## 18. Study Limitations

The following were some limitations of the study:This study is a narrative review, so a meta-analysis was not conducted.Studies in languages other than English could not be included.Articles that need to be accessed through institutional access could not be accessed.

## 19. Conclusions

Many clinical, epidemiological, and animal model studies have demonstrated the deleterious effect of Type 2 DM on the brain [22,23,24,48,49,50,51,84,85,86,87,88,89,90]. Cognitive impairment and a rise in the risk of vascular and Alzheimer’s dementia are the immediate negative impacts of Type 2 diabetes. Many pathologies in Type 2 DM result in neural damage and cognitive decline. Vascular, inflammatory, oxidative stress, and metabolic mechanisms contribute to the development of neuronal pathologies [Figure 5]. Studies indicate that the functional and structural integrity of the central nervous system is altered in Type 2 DM due to insulin excess or insulin resistance [147,148,149,150,151,152]. In Type 2 diabetes, glucose metabolism impairment, oxidative stress in cell organelles, and insulin resistance lead to increased production and secretion of Aβ. Hyperphosphorylation of Tau protein also results from dysregulation of various signaling cascades in Type 2 diabetes. Neuronal apoptosis synapse loss results from insulin resistance. The extent of neuronal damage is influenced by modifiers such as Apo E ε4 that promote the pathogenesis of AD in Type 2 diabetes. The interactive relationship between Type 2 diabetes and dementia is complex. Still, there is also potential for developing a therapy that may help prevent or halt the progress of dementia resulting from Type 2 diabetes [229].

Globally, AD is a leading cause of morbidity and mortality as available therapeutic options only address symptomatic issues. It has been reported that glucagon-like peptide-1 (GLP-1) receptor agonists such as liraglutide and semaglutide are effective glucose-lowering agents. Moreover, post-hoc analysis on the pooled data demonstrated a significant appraised hazard ratio of 0.47 (0.25; 0.86) 95%CI in favor of the GLP-1 RA medication versus placebo. These medications exhibit neurotrophic and neuroprotective effects, especially in AD cases [230,231,232,233,234].

## 20. Recommendations

More studies should be carried out further to understand the neuronal dysfunction mechanisms in Type 2 diabetes. Neuronal damage may be reduced using conventional approaches to maintain strict glycemic control with lifestyle modification and medication. Furthermore, interventions specific for pathways of pathologies and modifiers involved can be given to halt the progress of dementia. Evidence synthesis to produce a guideline for further research should be done to understand how the mechanisms of the disease process may be used to design effective interventions and bring about change. Progress in this field of research and finding ways to reverse cognitive impairment in diabetic patients will require the combined effort from neuropathologists, endocrinologists, neuropharmacologists, caregivers, and other health professionals.

## 21. Professional Explications

DM, a metabolic disorder, is suffered by millions worldwide. In 2014, there were 422 million individuals with DM, expected to rise to 693 million by 2045 [1,2,3]. DM’s chronic complications may develop and progress due to hyperglycemia and superoxide overproduction [4,5,6]. Pathways that lead to chronic complications of DM include protein kinase activation, advanced glycosylation end-product, inflammation, expression and action of cytokines, inflammatory mediators and hormones, polyol pathway, and increase in hexosamine activity [7,8]. The effects of DM and its complications on the central nervous system are not entirely understood. DM has been associated with cognitive decline with a raised risk of dementia (both vascular dementia and AD) [10]. Most diabetic subjects developing dementia are above the age of 65, but diabetes acts as a risk factor for dementia development before 65 years [19,20,21]. Studies, including original research, meta-analysis, and systemic review, have observed that the relative risk of dementia (vascular dementia and AD) were higher for diabetic individuals than non-diabetic subjects [22,23,32].

Among the types of dementia, vascular dementia results from blood flow reduction to the brain leading to the hampering of cognitive function. The suffering individual has anxiety, forgetfulness, depression, loss of working memory, reasoning, planning, task execution, and thinking [34,35]. A risk factor for vascular dementia is DM.

In vascular dementia, large and small blood vessels are involved, and microinfarct, lacunar infarct, macro infarct, micro- bleed, and changes in white matter have been noted in subjects. In addition, as indicated in neuropathological studies, Lacunar infarcts and microinfarcts are important risk factors for pure vascular dementia development [38,39,40,41,42]. Vascular lesions, leading to vascular dementia, result from atherosclerotic plaques in small cerebral vessels, lipohyalinosis in the vessel wall, microvasculature distortion, vessel wall stiffening, and complete integrity loss of blood vessels [44]. Lesions eventually cause demyelination, axon loss, vacuolation, and lacunar infarcts that damage the white matter and lead to a sharp decline, particularly in executive and motor function [48,49]. Studies of the brains of diabetic individuals have found silent brain infarcts, cerebral microbleeds, white matter lesions in MRI brain scans, and neuroimaging studies of brains of diabetic subjects showed lacunar infarcts and brown atrophy [24,25,50,51]. The pathological changes in vasculature in diabetic individuals can be attributed to the environment of inflammation created in this chronic metabolic condition [53]. Glycation of blood proteins due to hyperglycemia in DM cause the production of advanced glycation products in blood vessels of the peripheral and central nervous system. AGE accumulation causes vessel inflammation utilizing interaction between AGE and RAGE with upregulation of vascular cell adhesion molecule 1 (VCAM-1) and activation of NF kβ [60,61]. VCAM 1 enhances the adhesiveness of monocyte permeability of vasculature while production of NF kβ promotes proinflammatory and atherosclerotic changes in vascular endothelium and smooth muscle cells [60,61]. Atherosclerosis is promoted with eventual thrombus formation due to increased adhesion molecule expression in the endothelium, reduction in vasodilation, and inflammatory action. These changes ultimately may lead to cerebral infarction [50,67].

Studies observing the effect of increased blood glucose levels on astrocytes in humans noted significantly increased production of inflammatory cytokines like TNFα, IL1, IL4, IL6 utilizing STAT3 and NFkβ pathways of inflammation [91]. Brain endothelial penetrability is increased due to provocative, inflammatory agents. TNF α aggravates BBB permeability by acting directly on the endothelium and through the production of endothelin −1 and release of IL-1β from astrocyte [86,87]. The release of IL-1β may result in a reduced concentration of tight junction protein called occludin and therefore increase the permeability of BBB [86]. In a hyperglycemic state, glucose utilization occurs through Protein Kinase C and AGE pathways, which causes overproduction of superoxide [83]. Activation of Protein Kinase C leads to ZO 1 phosphorylation, tight junction disruption, and increased VEGF expression [95]. A study has found a correlation between increased BBB permeability with dementia development [27].

A significant cause for vascular oxidative stress in diabetes is NADPH oxidase [103]. AGE is formed in diabetic subjects, resulting in secretion of MMP-9 from endothelial cells and BDNF receptor TRKB cleavage, thus decreasing neurotrophin signaling [28]. Myelin sheath damage and demyelination of axons are some of the inflammatory states and oxidative stress [96,153]. Demyelination leads to disruption of the integrity of axons exposure to the damaging effects of free radicals and cytokines in the brain’s white matter [112,113]. Lesions of white matter affect the structure and function of the brain with a decrease in utilization of glucose by the frontal lobe [39,116] and disruption of brain connectivity [29,120,121]. Damage to myelin sheath may compromise skilled motor learning and neuroplasticity functions, thus leading to impairment of cognition [41].

AD (the most common form of dementia) is characterized by senile plaque and Neuro fibrillar tangles [11]. There is brain protein misfolding with deposition of extracellular amyloid plaque followed by neurofibrillary tangles deposition and neuronal death in the brain [127,128,129]. Transfer of Aβ to the periphery from CNS is inhibited by systemic inflammatory cytokines like TNFα, IL 6, and C reactive protein [150]. In turn, the accumulation of Aβ can cause the activation of microglia in the brain, which secrete inflammatory cytokines like IL6, IL1β, TNFα. These cytokines can bind to insulin receptors and activate IRS-1 serine kinase, which in turn phosphorylate IRS and thus cause alteration of insulin signaling in the brain [151,152]. Microglial proinflammatory cytokines promote oxidative stress that causes insulin signaling impairment, synapse loss, axonal mitochondrial transport reduction [153,155], fragmentation, and mitochondria dysfunction [156]. Dysfunction of mitochondria (linked with a metabolic syndrome like diabetes, insulin resistance, and obesity) has been an early change in AD [157]. In patients with diabetes and AD, there is misfolding and elevation of IAPP [166] and accumulation of increased quantity of Aβ, tau hyper-phosphorylation [167]. AD and diabetic patients share features of atrophy of the brain, cerebral glucose reduction, and insulin resistance in the CNS [168].

One of the prime features of AD is an alteration of glucose metabolism in mitochondria marked by insulin signaling impairment, receptor activity alteration, and reduction in glucose uptake [172]. Energy metabolism by mitochondria is modified due to immune cells infiltration from the periphery and microglia activation, causing initiation of a cascade of intracellular signaling pathways [173]. Impairment of oxidative phosphorylation is induced by inflammatory cytokines released from activated microglia. The activated microglia modulates astrocyte activity and deteriorates neuron integrity [174]. In the postmortem AD brain, inflammation in AD patients with impaired pyruvate dehydrogenase activity and elevated IL1β, TNFα, and IL6 levels were noted, implying a hamper of the TCA cycle of mitochondria [122]. A study on mouse hippocampal cell lines and neuronal cell culture has observed that TNF α exposure leads to decreased ATP formation and basal respiration [175]. Key enzymes like α ketoglutarate dehydrogenase, Pyruvate Dehydrogenase, enzymes of electron transport chain like cytochrome oxidase are all reduced in the AD brain [179]. There is disruption of the electron transport chain, increase in production of superoxide radicals and ROS with the hindrance of ATP generation due to interaction of Aβ, hyperphosphorylated tau protein with mitochondrial protein [183].

One of the modulators of the relationship between AD and Type 2 diabetes is Apolipoprotein E. Type 2 diabetic patients with the ApoE ε 4 genes have a higher risk of developing AD [218]. There is an increase in neurofibrillary tangles, formation of amyloid plaque, and cerebral amyloid angiopathy in this allele in Type 2 diabetic subjects with AD. Insulin degrading enzymes are lower in carriers of ApoE ε 4, which alter insulin signaling and clearance of Aβ in Type 2 diabetes and AD [219,220]. APOE4 impairs insulin signaling by interacting with insulin receptors. Studies have found downregulation of brain peroxisome proliferator-activated receptor γ and its coactivator PGC1α, glucose uptake reduction, and impaired insulin signaling in APOE4-TR mice [226,227,228]. Thus, suggesting genotype of ApoE worsens AD in Type 2 diabetic subjects [214]. Although the relationship between Type 2 DM and dementia is complex, there is potential for the development of a therapy that may help in prevention, slowing of progress, and even reversal of dementia resulting from Type 2 DM [229,235,236].

## Figures and Tables

**Figure 1 biology-11-00382-f001:**
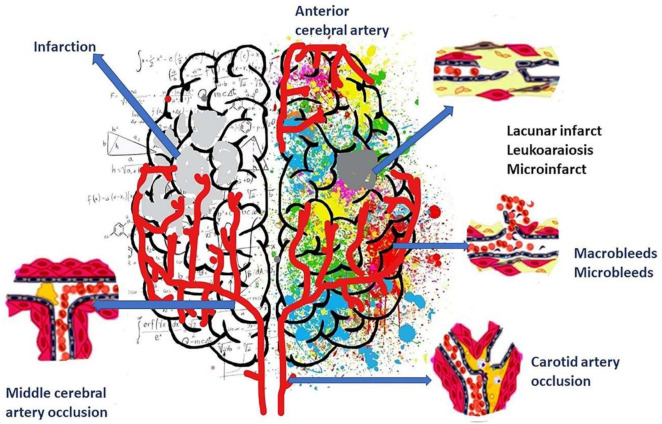
Microinfarct, Lacunar infarct, macro infarct, micro-bleed, hemorrhage, and white matter changes were observed in vascular dementia.

**Figure 2 biology-11-00382-f002:**
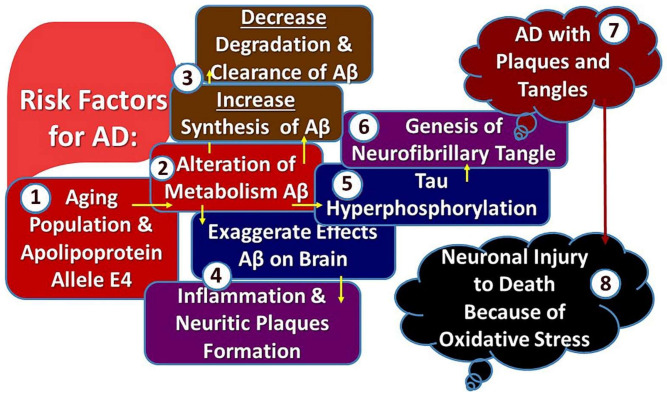
Illustrating Pathogenesis of Alzheimer’s Disease. Notes: Number within the figure denotes the sequences of Pathogenesis of AD.

**Figure 3 biology-11-00382-f003:**
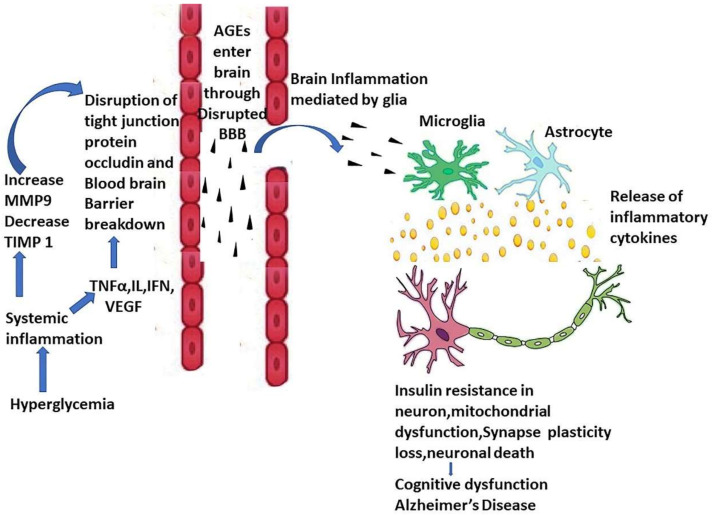
Systemic inflammation in DM causes a decrease of TIMP1 and an increase of MMP-9, increasing blood–brain barrier permeability. This triggers the release of inflammatory cytokines from microglia and astrocytes, which cause neuronal insulin resistance, mitochondrial dysfunction, neuronal death, and cognitive dysfunction. Notes: AGE: Advanced Glycation End product; MMP-9: Matrix Metalloprotease 9; TIMP 1: tissue inhibitor of metalloproteinases, TNFα: Tumor Necrosis Factor α, IL: Interleukins, INF: Interferons, VEGF: Vascular Endothelial Growth Factor, BBB: Blood Brain Barrier.

**Figure 4 biology-11-00382-f004:**
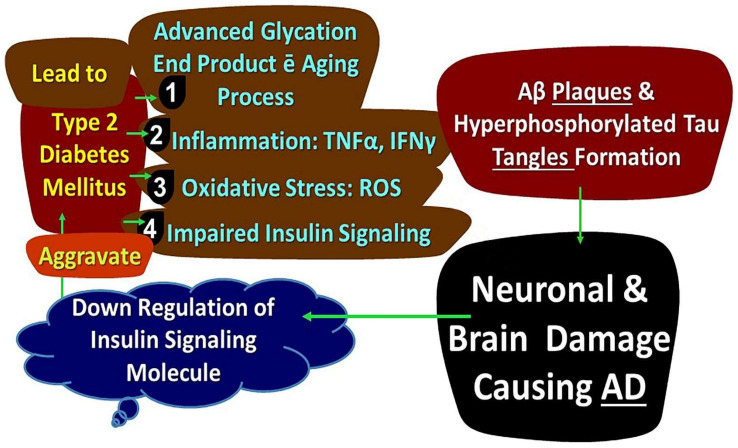
Illustrating Link between DM and Alzheimer’s Disease. Notes: AD: Alzheimer’s Disease, Aβ: Amyloid β.

**Figure 5 biology-11-00382-f005:**
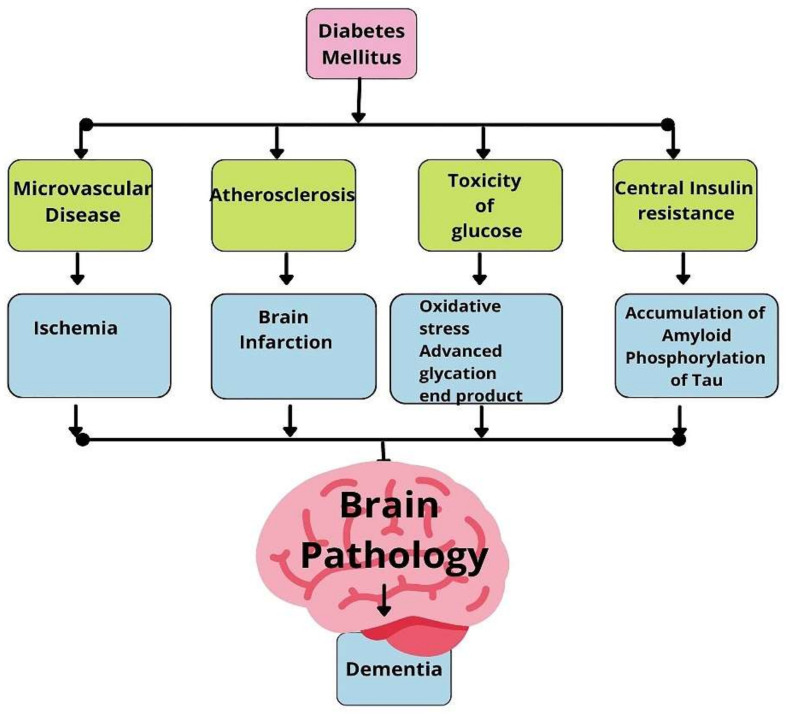
Mechanisms in Diabetes Mellitus which lead to Dementia.

## Data Availability

Not applicable.

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
