# Peer review of "Diabetes Mellitus: A Path to Amnesia, Personality, and Behavior Change"

_biology, 2022, doi:10.3390/biology11030382_

Round 1
Reviewer 1 Report
This review manuscript provides an overview of the evidence linking diabetes mellitus and dementia, and possible mechanisms contributing to dementia in diabetic patients.
The purpose of a review article is not a simple summary of literature in the field. Corresponding to every section of the manuscript, the authors should provide their unique insights. The recent summary of literature leads only to satisfaction but is not outstanding. The English language and style must be improved throughout the manuscript, and it needs extensive English editing. There is not any table to summarize the main results of previous human and animal studies.
- Page 2: ‘’Both AD and vascular dementia are the most common form of dementia.’’ Based on the reference, both AD and vascular dementia are two common types of dementia.
- Page 2: ‘’Several meta-analyses, systemic review and original studies have observed that the relative risk for all the types of dementia is 1.73(1.65-1.82) [20], for vascular dementia are 2.27 (1.94-2.66) [20], and for ADs 1.53 (1.42- 1.63) [21] for diabetic individuals when compared to nondiabetics.’’ Please paraphrase it.
- Figure 1 needs a more descriptive title and legend.
Page 3&4: ‘’Atherosclerotic plaques that affect small cerebral vessels, hyaline substance deposition in the vessel wall or lipohyalinosis, distortion of microvasculature, stiffening of the vessel wall due to complete fibrosis, loss of vessel wall integrity result in the vascular lesions that lead to Vascular Dementia [33].’’ Please paraphrase it for clarity.
Page 4: ‘’Thickened basement membrane in arterioles that become tortuous; reduced capillaries and non-functional capillaries with no endothelial cells and venules having collagen deposit have been noted [36].’’ Please paraphrase it.
Page 4: The section ‘’Diabetes Mellitus as a risk factor of Vascular Dementia’’ needs more evidence.
Page 4: ‘’In diabetic patient’s chronic hyperglycemia, hyperinsulinemia, dyslipidemia, and hypertension are risk factors for the formation of atherosclerosis [49].’’ Please paraphrase it.
Page 4: ‘’Reactive Oxidative Species production increases in Diabetes Mellitus [53]. A link between hyperglycemia and increased ROS formation was observed in studies carried out in vitro [54].’’ Please use a complete form of the word and its abbreviation in parenthesis (here ROS) when you use a word for the first time. Then, follow the consistency of abbreviation. Please check the entire manuscript for other words as well.
Page 5: ‘’In Vascular Cognitive Impairment, plasma protein albumin was noted to be increased in CSF, suggesting BBB breakdown [63].’’ Please avoid the abbreviation of ‘’CSF’’ and use full name for CSF, as there is not a full name for CSF in the manuscript.
Page 5: ‘’Activated RAGE increases the production of Reactive Oxygen Species with eventual activation of NF κβ, which promotes the release of inflammatory mediators [85-87].’’ Please use the abbreviation form of Reactive Oxygen Species.
- Please discuss if there is any link/association between dysfunctional adipose tissue and dementia, including AD and vascular dementia.
Page 6: In the section ‘’Diabetes Mellitus and AD’’, there is only a description of AD. Please justify it.
Page 6&7: Please provide a figure for section ‘’Pathogenesis of Alzheimer’s Disease’’ to show a proposed model of AD pathogenesis.
Page 7: ‘’Dysfunction of mitochondria-associated with a metabolic syndrome like diabetes, insulin resistance, and obesity has been an early change in AD (140).’’ Please discuss more the link between mitochondrial dysfunction, diabetes, and AD.
Page 8: ‘’AGE: Advanced Glycation End product; MMP 9: Matrix Metalloprotease 9; TIMP 1: tissue inhibitor of metalloproteinases.’’ Should be moved to figure 2 legend.
Page 10: ‘’Many clinical, epidemiological, and animal model studies have demonstrated the deleterious effect of Type 2 Diabetes Mellitus on the brain.’’ The references are missing.
Page 10: ‘’Studies indicate that the functional and structural integrity of the central nervous system is altered in Type 2 Diabetes Mellitus due to insulin excess or insulin resistance.’’ The references are missing.
Page 10: ‘’ Glucose metabolism is impaired, and oxidative stress in cell organelles in Type 2 diabetes: Aβ production and secretion increases due to insulin resistance.’’ Please paraphrase it.
Page 12: ‘’Avascular Dementia is Diabetes Mellitus [23,25-28].’’ Please paraphrase it.
Author Response
Reviewer I
Open Review
(x) I would not like to sign my review report
( ) I would like to sign my review report
English language and style
(x) Extensive editing of English language and style required
( ) Moderate English changes required
( ) English language and style are fine/minor spell check required
( ) I don't feel qualified to judge about the English language and style
|
Is the work a significant contribution to the field? |
|
|
This paper has been EDITED by an English Language Expert Is the work well organized and comprehensively described? |
|
|
Is the work scientifically sound and not misleading? |
|
|
Are there appropriate and adequate references to related and previous work? |
|
|
Is the English used correct and readable? |
Comments and Suggestions for Authors
This review manuscript provides an overview of the evidence linking diabetes mellitus and dementia, and possible mechanisms contributing to dementia in diabetic patients.
The purpose of a review article is not a simple summary of literature in the field. Corresponding to every section of the manuscript, the authors should provide their unique insights. The recent summary of literature leads only to satisfaction but is not outstanding. The English language and style must be improved throughout the manuscript, and it needs extensive English editing. There is not any table to summarize the main results of previous human and animal studies.
Many Thanks. We have added a Table as per your expert advice.
- Page 2: ‘’Both AD and vascular dementia are the most common form of dementia.’’ Based on the reference, both AD and vascular dementia are two common types of dementia.
Thanks Sir. We have corrected accordingly.
- Page 2: ‘’Several meta-analyses, systemic review and original studies have observed that the relative risk for all the types of dementia is 1.73(1.65-1.82) [20], for vascular dementia are 2.27 (1.94-2.66) [20], and for ADs 1.53 (1.42- 1.63) [21] for diabetic individuals when compared to nondiabetics.’’ Please paraphrase it for clarity.
We have paraphrase it to clarify.
- Figure 1 needs a more descriptive title and legend.
We have taken care of the Issue: title and legend
Page 3&4: ‘’Atherosclerotic plaques that affect small cerebral vessels, hyaline substance deposition in the vessel wall or lipohyalinosis, distortion of microvasculature, stiffening of the vessel wall due to complete fibrosis, loss of vessel wall integrity result in the vascular lesions that lead to Vascular Dementia [33].’’ Please paraphrase it for clarity.
We have paraphrase it to clarify.
Page 4: ‘’Thickened basement membrane in arterioles that become tortuous; reduced capillaries and non-functional capillaries with no endothelial cells and venules having collagen deposit have been noted [36].’’ Please paraphrase it.
We have paraphrase it to clarify.
Page 4: The section ‘’Diabetes Mellitus as a risk factor of Vascular Dementia’’ needs more evidence.
We added and discussed as per your kind advice.
Page 4: ‘’In diabetic patient’s chronic hyperglycemia, hyperinsulinemia, dyslipidemia, and hypertension are risk factors for the formation of atherosclerosis [49].’’ Please paraphrase it.
We have paraphrase it to clarify.
Page 4: ‘’Reactive Oxidative Species production increases in Diabetes Mellitus [53]. A link between hyperglycemia and increased ROS formation was observed in studies carried out in vitro [54].’’ Please use a complete form of the word and its abbreviation in parenthesis (here ROS) when you use a word for the first time. Then, follow the consistency of abbreviation. Please check the entire manuscript for other words as well.
We have altered as per your expert opinion.
Page 5: ‘’In Vascular Cognitive Impairment, plasma protein albumin was noted to be increased in CSF, suggesting BBB breakdown [63].’’ Please avoid the abbreviation of ‘’CSF’’ and use full name for CSF, as there is not a full name for CSF in the manuscript.
We have altered as per your expert opinion the full form of CSF added.
Page 5: ‘’Activated RAGE increases the production of Reactive Oxygen Species with eventual activation of NF κβ, which promotes the release of inflammatory mediators [85-87].’’ Please use the abbreviation form of Reactive Oxygen Species.
We have altered as per your expert opinion.
- Please discuss if there is any link/association between dysfunctional adipose tissue and dementia, including AD and vascular dementia.
We have altered as per your expert advice.
Page 6: In the section ‘’Diabetes Mellitus and AD’’, there is only a description of AD. Please justify it.
We have altered as per your expert advice. The title of the section has been corrected. We are extremely sorry for the error.
Page 6&7: Please provide a figure for section ‘’Pathogenesis of Alzheimer’s Disease’’ to show a proposed model of AD pathogenesis.
Thanks Sir. A New Figure Added with this Paper
Page 7: ‘’Dysfunction of mitochondria-associated with a metabolic syndrome like diabetes, insulin resistance, and obesity has been an early change in AD (140).’’ Please discuss more the link between mitochondrial dysfunction, diabetes, and AD.
Thanks Sir. We have altered as per your expert advice.
Page 8: ‘’AGE: Advanced Glycation End product; MMP 9: Matrix Metalloprotease 9; TIMP 1: tissue inhibitor of metalloproteinases.’’ Should be moved to figure 2 legend.
Thanks Sir We altered as per your kind advice.
Page 10: ‘’Many clinical, epidemiological, and animal model studies have demonstrated the deleterious effect of Type 2 Diabetes Mellitus on the brain.’’ The references are missing.
A reference included as per your kind advice.
Page 10: ‘’Studies indicate that the functional and structural integrity of the central nervous system is altered in Type 2 Diabetes Mellitus due to insulin excess or insulin resistance.’’ The references are missing.
A reference included as per your kind advice.
Page 10: ‘’ Glucose metabolism is impaired, and oxidative stress in cell organelles in Type 2 diabetes: Aβ production and secretion increases due to insulin resistance.’’ Please paraphrase it.
We have paraphrased this section.
Page 12: ‘’Avascular Dementia is Diabetes Mellitus [23,25-28].’’ Please paraphrase it.
We have paraphrased this section.
Submission Date: 13 January 2022. Date of this review: 17 Jan 2022 17:33:56
Reviewer II
Open Review
( ) I would not like to sign my review report
(x) I would like to sign my review report
English language and style
( ) Extensive editing of English language and style required
( ) Moderate English changes required
(x) English language and style are fine/minor spell check required
( ) I don't feel qualified to judge about the English language and style
Is the work a significant contribution to the field?
Is the work well organized and comprehensively described?
Is the work scientifically sound and not misleading?
Are there appropriate and adequate references to related and previous work?
Is the English used correct and readable?
Comments and Suggestions for Authors
This is a very good revision. I have some comments about the paper:
In the fifth paragraph of page 2 is wrote 'several meta-analyses, systemic review and original studies...", the correct is sytemic or systematic review?
Many Thanks Sir. We have altered.
In page 6, second paragraph of Diabetes Mellitus and AD, the citation number 109 seems to be in the wrong place.
Many Thanks. We have updated 109 reference.
The manuscript is long, however has the necessary information.
Many Thanks Sir. Just for kind information Publishers do not have word count limitations.
In reference number 168 the correct name of the journal is Brain Res Rev.
Thanks Sir. We have Corrected.
Pages of the articles in the reference list do not have the same presentations. In the reference 42 the pages were "4036-42" and in reference 43 "1629-1648". Other references have the same problem.
We have checked and corrected. Moreover, to my best knowledge it automatically formatted by publisher.
Submission Date: 13 January 2022. Date of this review: 26 Jan 2022 14:51:26.
Reviewer III
Open Review
( ) I would not like to sign my review report
(x) I would like to sign my review report
English language and style
( ) Extensive editing of English language and style required
( ) Moderate English changes required
(x) English language and style are fine/minor spell check required
( ) I don't feel qualified to judge about the English language and style
Is the work a significant contribution to the field?
Is the work well organized and comprehensively described?
Is the work scientifically sound and not misleading?
Are there appropriate and adequate references to related and previous work?
Is the English used correct and readable?
Comments and Suggestions for Authors
The following review describes how diabetes and hyperglycemia and overproduction of superoxide induces the development and progression of chronic complications of Diabetes Mellitus related with cognition. In both vascular and Alzheimer dementia, cytokines may cause macrophage activation, thus inducing a cascade of pro-inflammatory changes and finally changes in BBB permeability.
Despite the review is very interesting, sometimes it is difficult to follow due to a concentration of the information. Simplification of the content would facilitate the reader comprehension or inclusion of schemes and take-home message. Increase the details of Figure 2 (include the sequential pathways involved i.e. TNF, IL-1 - - - -Occludin, VEGF). Create a similar picture related with Diabetes and AD. Create a final take-home message.
Many Thanks Sir. We have added a New Figure.
Correct the sentence: In Type 2 Diabetes Mellitus, primary sources for inflammatory cytokines like IL-1β, IL 6, and TNF α are activated macrophages found in adipose tissue.
Many Thanks. We have corrected as advised.
Correction: Damage of myeline sheath and demyelination.
Thanks Sir. We Have altered.
Abrev: NADPH (Nicotinamide adenine dinucleotide phosphate hydrogen) the NADPH appears before this line.
We have corrected.
Discussion: Include the information of some clinical trials using drugs for Diabetis for AD treatment: Semaglutide, in clinical trial EVOKE. Semaglutide, a GLP1 drug, has shown to decrease risk of dementia in real world data and post hoc analysis.
Thanks Sir. We have incorporated your advice in Conclusion section.
Submission Date: 13 January 2022. Date of this review: 08 Feb 2022 17:13:38
© 1996-2022 MDPI (Basel, Switzerland) unless otherwise stated
Reviewer IV
Open Review
( ) I would not like to sign my review report
(x) I would like to sign my review report
English language and style
( ) Extensive editing of English language and style required
( ) Moderate English changes required
( ) English language and style are fine/minor spell check required
(x) I don't feel qualified to judge about the English language and style
|
Is the work a significant contribution to the field? |
|
|
Is the work well organized and comprehensively described? |
|
|
Is the work scientifically sound and not misleading? |
|
|
Are there appropriate and adequate references to related and previous work? |
|
|
Is the English used correct and readable? |
Comments and Suggestions for Authors
The manuscript is very interesting as it summarizes the current knowledge on the association between type 2 diabetes mellitus, Alzheimer's disease, inflammation, oxidative stress and cognitive impairment, as well as the biochemical and molecular mechanisms involved.
Small changes are necessary to improve the manuscript:
Page 6: The text states that "NADPH oxidase is an important cause of vascular oxidative stress......". It is not the only cause, the activation or increase of its enzymatic activity would be the cause, its function is the generation of superoxide anion. Indicate this situation and explain the function of this enzyme complex.
Many Thanks Sir. We have incorporated your ideas.
Page 7: The full name of the NADPH oxidase enzyme should be indicated on page 6, which is where it is first cited. Page 7: Indicate which are specifically the reactive oxygen species involved. Pay attention that the word insulin is sometimes written in uppercase and other times in lowercase. Page 9: Change "Kreb" por "Krebs" because this is the correct name.
Many Thanks Sir. We have altered.
At the end of the manuscript, there is a section named: Expert opinion. It is not clear why these paragraphs were added as they are already included in the manuscript.
Thanks, Sir. Many publishers have the option of such a section as mandatory. As publishers do not have any restriction of word limit. Thereby we would like to retain.
Submission Date: 13 January 2022. Date of this review. 04 Feb 2022 17:55:43

Reviewer 2 Report
This is a very good revision. I have some comments about the paper:
- In the fifth paragraph of page 2 is wrote 'several meta-analyses, systemic review and original studies...", the correct is sytemic or systematic review?
- In page 6, second paragraph of Diabetes Mellitus and AD, the citation number 109 seems to be in the wrong place.
- The manuscript is long, however has the necessary informations.
- In reference number 168 the correct name of the journal is Brain Res Rev.
- Pages of the articles in the reference list do not have the same presentations. In the reference 42 the pages were "4036-42" and in reference 43 "1629-1648". Other references have the same problem.
Author Response
Reviewer I
Open Review
(x) I would not like to sign my review report
( ) I would like to sign my review report
English language and style
(x) Extensive editing of English language and style required
( ) Moderate English changes required
( ) English language and style are fine/minor spell check required
( ) I don't feel qualified to judge about the English language and style
|
Is the work a significant contribution to the field? |
|
|
This paper has been EDITED by an English Language Expert Is the work well organized and comprehensively described? |
|
|
Is the work scientifically sound and not misleading? |
|
|
Are there appropriate and adequate references to related and previous work? |
|
|
Is the English used correct and readable? |
Comments and Suggestions for Authors
This review manuscript provides an overview of the evidence linking diabetes mellitus and dementia, and possible mechanisms contributing to dementia in diabetic patients.
The purpose of a review article is not a simple summary of literature in the field. Corresponding to every section of the manuscript, the authors should provide their unique insights. The recent summary of literature leads only to satisfaction but is not outstanding. The English language and style must be improved throughout the manuscript, and it needs extensive English editing. There is not any table to summarize the main results of previous human and animal studies.
Many Thanks. We have added a Table as per your expert advice.
- Page 2: ‘’Both AD and vascular dementia are the most common form of dementia.’’ Based on the reference, both AD and vascular dementia are two common types of dementia.
Thanks Sir. We have corrected accordingly.
- Page 2: ‘’Several meta-analyses, systemic review and original studies have observed that the relative risk for all the types of dementia is 1.73(1.65-1.82) [20], for vascular dementia are 2.27 (1.94-2.66) [20], and for ADs 1.53 (1.42- 1.63) [21] for diabetic individuals when compared to nondiabetics.’’ Please paraphrase it for clarity.
We have paraphrase it to clarify.
- Figure 1 needs a more descriptive title and legend.
We have taken care of the Issue: title and legend
Page 3&4: ‘’Atherosclerotic plaques that affect small cerebral vessels, hyaline substance deposition in the vessel wall or lipohyalinosis, distortion of microvasculature, stiffening of the vessel wall due to complete fibrosis, loss of vessel wall integrity result in the vascular lesions that lead to Vascular Dementia [33].’’ Please paraphrase it for clarity.
We have paraphrase it to clarify.
Page 4: ‘’Thickened basement membrane in arterioles that become tortuous; reduced capillaries and non-functional capillaries with no endothelial cells and venules having collagen deposit have been noted [36].’’ Please paraphrase it.
We have paraphrase it to clarify.
Page 4: The section ‘’Diabetes Mellitus as a risk factor of Vascular Dementia’’ needs more evidence.
We added and discussed as per your kind advice.
Page 4: ‘’In diabetic patient’s chronic hyperglycemia, hyperinsulinemia, dyslipidemia, and hypertension are risk factors for the formation of atherosclerosis [49].’’ Please paraphrase it.
We have paraphrase it to clarify.
Page 4: ‘’Reactive Oxidative Species production increases in Diabetes Mellitus [53]. A link between hyperglycemia and increased ROS formation was observed in studies carried out in vitro [54].’’ Please use a complete form of the word and its abbreviation in parenthesis (here ROS) when you use a word for the first time. Then, follow the consistency of abbreviation. Please check the entire manuscript for other words as well.
We have altered as per your expert opinion.
Page 5: ‘’In Vascular Cognitive Impairment, plasma protein albumin was noted to be increased in CSF, suggesting BBB breakdown [63].’’ Please avoid the abbreviation of ‘’CSF’’ and use full name for CSF, as there is not a full name for CSF in the manuscript.
We have altered as per your expert opinion the full form of CSF added.
Page 5: ‘’Activated RAGE increases the production of Reactive Oxygen Species with eventual activation of NF κβ, which promotes the release of inflammatory mediators [85-87].’’ Please use the abbreviation form of Reactive Oxygen Species.
We have altered as per your expert opinion.
- Please discuss if there is any link/association between dysfunctional adipose tissue and dementia, including AD and vascular dementia.
We have altered as per your expert advice.
Page 6: In the section ‘’Diabetes Mellitus and AD’’, there is only a description of AD. Please justify it.
We have altered as per your expert advice. The title of the section has been corrected. We are extremely sorry for the error.
Page 6&7: Please provide a figure for section ‘’Pathogenesis of Alzheimer’s Disease’’ to show a proposed model of AD pathogenesis.
Thanks Sir. A New Figure Added with this Paper
Page 7: ‘’Dysfunction of mitochondria-associated with a metabolic syndrome like diabetes, insulin resistance, and obesity has been an early change in AD (140).’’ Please discuss more the link between mitochondrial dysfunction, diabetes, and AD.
Thanks Sir. We have altered as per your expert advice.
Page 8: ‘’AGE: Advanced Glycation End product; MMP 9: Matrix Metalloprotease 9; TIMP 1: tissue inhibitor of metalloproteinases.’’ Should be moved to figure 2 legend.
Thanks Sir We altered as per your kind advice.
Page 10: ‘’Many clinical, epidemiological, and animal model studies have demonstrated the deleterious effect of Type 2 Diabetes Mellitus on the brain.’’ The references are missing.
A reference included as per your kind advice.
Page 10: ‘’Studies indicate that the functional and structural integrity of the central nervous system is altered in Type 2 Diabetes Mellitus due to insulin excess or insulin resistance.’’ The references are missing.
A reference included as per your kind advice.
Page 10: ‘’ Glucose metabolism is impaired, and oxidative stress in cell organelles in Type 2 diabetes: Aβ production and secretion increases due to insulin resistance.’’ Please paraphrase it.
We have paraphrased this section.
Page 12: ‘’Avascular Dementia is Diabetes Mellitus [23,25-28].’’ Please paraphrase it.
We have paraphrased this section.
Submission Date: 13 January 2022. Date of this review: 17 Jan 2022 17:33:56
Reviewer II
Open Review
( ) I would not like to sign my review report
(x) I would like to sign my review report
English language and style
( ) Extensive editing of English language and style required
( ) Moderate English changes required
(x) English language and style are fine/minor spell check required
( ) I don't feel qualified to judge about the English language and style
Is the work a significant contribution to the field?
Is the work well organized and comprehensively described?
Is the work scientifically sound and not misleading?
Are there appropriate and adequate references to related and previous work?
Is the English used correct and readable?
Comments and Suggestions for Authors
This is a very good revision. I have some comments about the paper:
In the fifth paragraph of page 2 is wrote 'several meta-analyses, systemic review and original studies...", the correct is sytemic or systematic review?
Many Thanks Sir. We have altered.
In page 6, second paragraph of Diabetes Mellitus and AD, the citation number 109 seems to be in the wrong place.
Many Thanks. We have updated 109 reference.
The manuscript is long, however has the necessary information.
Many Thanks Sir. Just for kind information Publishers do not have word count limitations.
In reference number 168 the correct name of the journal is Brain Res Rev.
Thanks Sir. We have Corrected.
Pages of the articles in the reference list do not have the same presentations. In the reference 42 the pages were "4036-42" and in reference 43 "1629-1648". Other references have the same problem.
We have checked and corrected. Moreover, to my best knowledge it automatically formatted by publisher.
Submission Date: 13 January 2022. Date of this review: 26 Jan 2022 14:51:26.
Reviewer III
Open Review
( ) I would not like to sign my review report
(x) I would like to sign my review report
English language and style
( ) Extensive editing of English language and style required
( ) Moderate English changes required
(x) English language and style are fine/minor spell check required
( ) I don't feel qualified to judge about the English language and style
Is the work a significant contribution to the field?
Is the work well organized and comprehensively described?
Is the work scientifically sound and not misleading?
Are there appropriate and adequate references to related and previous work?
Is the English used correct and readable?
Comments and Suggestions for Authors
The following review describes how diabetes and hyperglycemia and overproduction of superoxide induces the development and progression of chronic complications of Diabetes Mellitus related with cognition. In both vascular and Alzheimer dementia, cytokines may cause macrophage activation, thus inducing a cascade of pro-inflammatory changes and finally changes in BBB permeability.
Despite the review is very interesting, sometimes it is difficult to follow due to a concentration of the information. Simplification of the content would facilitate the reader comprehension or inclusion of schemes and take-home message. Increase the details of Figure 2 (include the sequential pathways involved i.e. TNF, IL-1 - - - -Occludin, VEGF). Create a similar picture related with Diabetes and AD. Create a final take-home message.
Many Thanks Sir. We have added a New Figure.
Correct the sentence: In Type 2 Diabetes Mellitus, primary sources for inflammatory cytokines like IL-1β, IL 6, and TNF α are activated macrophages found in adipose tissue.
Many Thanks. We have corrected as advised.
Correction: Damage of myeline sheath and demyelination.
Thanks Sir. We Have altered.
Abrev: NADPH (Nicotinamide adenine dinucleotide phosphate hydrogen) the NADPH appears before this line.
We have corrected.
Discussion: Include the information of some clinical trials using drugs for Diabetis for AD treatment: Semaglutide, in clinical trial EVOKE. Semaglutide, a GLP1 drug, has shown to decrease risk of dementia in real world data and post hoc analysis.
Thanks Sir. We have incorporated your advice in Conclusion section.
Submission Date: 13 January 2022. Date of this review: 08 Feb 2022 17:13:38
© 1996-2022 MDPI (Basel, Switzerland) unless otherwise stated
Reviewer IV
Open Review
( ) I would not like to sign my review report
(x) I would like to sign my review report
English language and style
( ) Extensive editing of English language and style required
( ) Moderate English changes required
( ) English language and style are fine/minor spell check required
(x) I don't feel qualified to judge about the English language and style
|
Is the work a significant contribution to the field? |
|
|
Is the work well organized and comprehensively described? |
|
|
Is the work scientifically sound and not misleading? |
|
|
Are there appropriate and adequate references to related and previous work? |
|
|
Is the English used correct and readable? |
Comments and Suggestions for Authors
The manuscript is very interesting as it summarizes the current knowledge on the association between type 2 diabetes mellitus, Alzheimer's disease, inflammation, oxidative stress and cognitive impairment, as well as the biochemical and molecular mechanisms involved.
Small changes are necessary to improve the manuscript:
Page 6: The text states that "NADPH oxidase is an important cause of vascular oxidative stress......". It is not the only cause, the activation or increase of its enzymatic activity would be the cause, its function is the generation of superoxide anion. Indicate this situation and explain the function of this enzyme complex.
Many Thanks Sir. We have incorporated your ideas.
Page 7: The full name of the NADPH oxidase enzyme should be indicated on page 6, which is where it is first cited. Page 7: Indicate which are specifically the reactive oxygen species involved. Pay attention that the word insulin is sometimes written in uppercase and other times in lowercase. Page 9: Change "Kreb" por "Krebs" because this is the correct name.
Many Thanks Sir. We have altered.
At the end of the manuscript there is a section named: Expert opinion. It is not clear why these paragraphs were added as they are already included in the manuscript.
Thanks Sir. Many publishers have option of such section as mandatory. As publishers does not any restriction of word limit. Thereby we would like to retain.
Submission Date: 13 January 2022. Date of this review. 04 Feb 2022 17:55:43

Reviewer 3 Report
The following review describes how diabetes and hyperglycemia and overproduction of superoxide induces the development and progression of chronic complications of Diabetes Mellitus related with cognition. In both vascular and Alzheimer dementia, cytokines may cause macrophage activation, thus inducing a cascade of pro-inflammatory changes and finally changes in BBB permeability.
Despite the review is very interesting, sometimes it is difficult to follow due to a concentration of the information. Simplification of the content would facilitate the reader comprehension or inclusion of schemes and take-home message. Increase the details of Figure 2 (include the sequential pathways involved i.e. TNF, IL-1 - - - -Occludin, VEGF). Create a similar picture related with Diabetis and AD. Create a final take-home message.
Correct the sentence: In Type 2 Diabetes Mellitus, primary sources for inflammatory cytokines like IL-1β, IL 6, and TNF α are activated macrophages found in adipose tissue.
Correction: Damage of myeline sheath and demyelination
Abrev: NADPH (Nicotinamide adenine dinucleotide phosphate hydrogen) the NADPH appears before this line.
Discussion: Include the information of some clinical trials using drugs for Diabetis for AD treatment: Semaglutide, in clinical trial EVOKE. Semaglutide, a GLP1 drug, has shown to decrease risk of dementia in real world data and post hoc analysis.
Author Response

(The authors gave the same response as above.)

Reviewer 4 Report
The manuscript is very interesting as it summarizes the current knowledge on the association between type 2 diabetes mellitus, Alzheimer's disease, inflammation, oxidative stress and cognitive impairment, as well as the biochemical and molecular mechanisms involved.Small changes are necessary to improve the manuscript:
Page 6:The text states that "NADPH oxidase is an important cause of vascular oxidative stress......". It is not the only cause, the activation or increase of its enzymatic activity would be the cause, its function is the generation of superoxide anion. Indicate this situation and explain the function of this enzyme complex.
Page 7:The full name of the NADPH oxidase enzyme should be indicated on page 6, which is where it is first cited. Page 7: Indicate which are specifically the reactive oxygen species involved. Pay attention that the word insulin is sometimes written in uppercase and other times in lowercase. Page 9: Change "Kreb" por "Krebs" because this is the correct name.
At the end of the manuscript there is a section named: Expert opinion. It is not clear why these paragraphs were added as they are already included in the manuscript.
Author Response

(The authors gave the same response as above.)

Round 2
Reviewer 1 Report
- Please check the entire manuscript for typos, punctuation, and grammatical errors.
- Please revise the table as it is long and difficult to follow. The ‘’results’’ part of the table also needs to be summarized.
- Page 2: ‘’Both AD and vascular dementia are the most common form of dementia.’’ - Based on the reference of (16), both AD and vascular dementia are two common types of dementia. Please justify it.
- Page 2: ‘’The relative risk for dementia was noted to be 1.73(1.65-1.82) and for vascular dementia to be 2.27(1.94-2.66) for Diabetic subjects in a meta-analysis of prospective observational studies on DM and risk of dementia [20]. Another meta-analysis of cohort studies on DM and risk of Alzheimer's disease observed Relative risk for AD 1.53 (1.42-1.63) for Diabetic individuals [21].’’ - Please paraphrase it.
- Page 6: ‘’…distortion of microvasculature; vessel wall stiffening due to fibrosis; vessel wall integrity loss [42].’’ - Please put and before ‘’vessel wall integrity loss [42]’’.
- Figure 2 is not clear for a reader to understand a proposed model of AD pathogenesis. Please revise it.
Author Response
Reviewer I
Open Review
(x) I would not like to sign my review report
( ) I would like to sign my review report
English language and style
( ) Extensive editing of English language and style required
(x) Moderate English changes required
( ) English language and style are fine/minor spell check required
( ) I don't feel qualified to judge about the English language and style
|
Is the work a significant contribution to the field? |
|
|
Is the work well organized and comprehensively described? |
|
|
Is the work scientifically sound and not misleading? |
|
|
Are there appropriate and adequate references to related and previous work? |
|
|
Is the English used correct and readable? |
Comments and Suggestions for Authors
- Please check the entire manuscript for typos, punctuation, and grammatical errors.
Many Thanks. We have taken care of the English Language.
- Please revise the table as it is long and difficult to follow. The ‘’results’’ part of the table also needs to be summarized.
Thanks Sir. We minimized Table Results Section.
- Page 2: ‘’Both AD and vascular dementia are the most common form of dementia.’’ - Based on the reference of (16), both AD and vascular dementia are two common types of dementia. Please justify it.
Thanks. We give effort to justify our statement.
- Page 2: ‘’The relative risk for dementia was noted to be 1.73(1.65-1.82) and for vascular dementia to be 2.27(1.94-2.66) for Diabetic subjects in a meta-analysis of prospective observational studies on DM and risk of dementia [20]. Another meta-analysis of cohort studies on DM and risk of Alzheimer's disease observed Relative risk for AD 1.53 (1.42-1.63) for Diabetic individuals [21].’’ - Please paraphrase it.
Thanks for expert comment. We have paraphrase as advice.
- Page 6: ‘’…distortion of microvasculature; vessel wall stiffening due to fibrosis; vessel wall integrity loss [42].’’ - Please put and before ‘’vessel wall integrity loss [42]’’.
Thanks. We have altered the mentioned sentence.
- Figure 2 is not clear for a reader to understand a proposed model of AD pathogenesis. Please revise it.
Submission Date
13 January 2022
Date of this review
15 Feb 2022 22:01:07
